# Oxidative stress changes interactions between 2 bacterial species from competitive to facilitative

**Rita Di Martino**[1]☉, **Aurore Picot**[1,2]☉, **Sara Mitri**☉[1]*

**1** Department of Fundamental Microbiology, University of Lausanne, Lausanne, Switzerland, **2** Center for Interdisciplinary Research in Biology (CIRB), Collège de France, CNRS, INSERM, Université PSL, Paris, France

☉ These authors contributed equally to this work.
* sara.mitri@unil.ch

**Data Availability Statement:** All raw data and computer code are available via Zenodo at the following DOIs: 10.5281/zenodo.8033844 and 10.5281/zenodo.10396268 respectively.

## Abstract

Knowing how species interact within microbial communities is crucial to predicting and controlling community dynamics, but interactions can depend on environmental conditions. The stress-gradient hypothesis (SGH) predicts that species are more likely to facilitate each other in harsher environments. Even if the SGH gives some intuition, quantitative modeling of the context-dependency of interactions requires understanding the mechanisms behind the SGH. In this study, we show with both experiments and a theoretical analysis that varying the concentration of a single compound, linoleic acid (LA), modifies the interaction between 2 bacterial species, *Agrobacterium tumefaciens* and *Comamonas testosteroni*, from competitive at a low concentration, to facilitative at higher concentrations where LA becomes toxic for one of the 2 species. We demonstrate that the mechanism behind facilitation is that one species is able to reduce reactive oxygen species (ROS) that are produced spontaneously at higher concentrations of LA, allowing for short-term rescue of the species that is sensitive to ROS and longer coexistence in serial transfers. In our system, competition and facilitation between species can occur simultaneously, and changing the concentration of a single compound can alter the balance between the two.

## Introduction

Multispecies microbial communities colonize almost every environment. Despite their small individual size, microbes can form very large populations that significantly affect their surroundings. For example, they can greatly influence the health and behavior of their living hosts [1]—for better or worse—or alter the physical and chemical properties of surfaces they are living on [2]. How these effects play out depends on a community's species composition and how it changes over time, which in turn depends on how the different species interact: who restricts or enhances whose growth. But how we expect microbial species to interact with one another, what drives their interactions, and how interactions shape long-term coexistence remain matters of debate [3–10].

**Funding:** RDM was funded by H2020 European Research Council grant 715097, AP by Swiss National Science Foundation Eccellenza grant PCEGP3_181272, and SM by both grants as well as the National Center of Competence in Research Microbiomes grant SNF 51NF40_180575. The funders had no role in study design, data collection and analysis, decision to publish, or preparation of the manuscript.

**Competing interests:** The authors have declared that no competing interests exist.

**Abbreviations:** AUC, area under the growth curve; LA, linoleic acid; LB, lysogeny broth; MM, minimal medium; MWF, metalworking fluid; ODE, ordinary differential equation; ROS, reactive oxygen species; SGH, stress-gradient hypothesis; TBA, thiobarbituric acid; TBARS, thiobarbituric acid reactive substance; TBHQ, tert-butylhydroquinone; TSA, tryptic soy agar; TSB, tryptic soy broth.

Here, we define interactions between species as the effect of one species on the growth and death of another and focus on environmentally mediated interactions [11]. A species may negatively affect another by making the environment worse for it, for example, by depleting a common resource or releasing a toxic waste product. Alternatively, a species may affect another positively by improving the environment for the other, or for itself, accidentally benefiting the other [11,12]. Such positive effects can happen through detoxification, cross-feeding, or the release of public goods such as siderophores [13,14].

Whether or not a species improves the environment or makes it more difficult for another species to grow will depend on the properties of the environment itself, suggesting that interactions should be context dependent [12,15]. Indeed, several studies have demonstrated that changes to the chemical composition of the environment can alter interaction sign [16–23].

Ultimately, our understanding of interspecies interactions should allow us to predict the long-term composition of microbial communities, i.e., which species are likely to coexist. Coexistence theory usually considers competitive interactions, although recent approaches also integrate facilitation and mutualism [24]. In particular, Modern Coexistence Theory [25,26] focuses on species–species interactions often with a general Lotka–Volterra framework, while Contemporary Niche Theory [27,28] sees interspecific interactions as mediated by an environment from a consumer–resource perspective [29], which takes the dependency of interactions on environmental conditions into account [11,30]. Indeed, coexistence outcomes can critically depend on the mechanistic details of interactions [24,31,32].

One example where context-dependency has been proposed is the stress-gradient hypothesis (SGH), which states that positive interactions between species are likely to increase with environmental stress. The SGH was originally described in plants [33,34], but has also been observed in bacterial communities [16,21,23,35]. In our previous work [23], we measured pairwise interactions between 4 bacterial species that were isolated from polluting industrial liquids called metalworking fluids (MWFs). We showed that all pairwise interactions were positive in toxic MWF, but became more competitive when the environment was made more benign. Although this result supports the SGH, the complexity of the MWF medium and its unknown chemical composition prevented us from identifying a mechanism to explain facilitation in this toxic environment.

In this study, we tested the SGH in a simpler system, combining experiments and mathematical models to study species interactions and coexistence in environmentally defined conditions. Our system consists of 2 bacterial species originally isolated from MWF, *Agrobacterium tumefaciens* (henceforth *At*) and *Comamonas testosteroni* (*Ct*), growing in a defined medium containing linoleic acid (LA), a fatty acid commonly present in many MWFs [36] as the sole carbon source. We chose this carbon source as it becomes toxic for *At*—but not *Ct*—at high concentration, making it suitable to explore the relationship between toxicity and interactions, and to mechanistically test the SGH and how facilitation affects coexistence.

We found that the toxicity of LA for *At* was due to the spontaneous accumulation of reactive oxygen species (ROS). Using the models, we show how to drive the interactions in the two-species co-culture towards more competition or facilitation by simply reducing or increasing the initial LA concentration, respectively, in line with SGH. Moreover, by artificially reducing environmental toxicity (by adding an antioxidant) at high LA concentration, we were also able to revert back to competition, hence mechanistically showing how the SGH works in our system. We then used our model to predict how interactions would affect long-term coexistence and found that toxicity extends the duration of coexistence in the short term, which was experimentally validated. Overall, the simplicity of our system allowed us to identify the mechanism behind the interactions between our bacteria and to shape them just by manipulating toxicity through the concentration of a single chemical compound.

## Results

### Linoleic acid has concentration-dependent effects in mono-culture

To build on our previous work [23] and track how pair-wise interactions change with varying compound concentration and varying toxicity, we designed a simpler medium containing a single MWF compound at a time. We selected 10 compounds commonly found in MWF and tested their effect on *At* and *Ct* in mono-culture at different concentrations (Fig 1A). As *Ct* was facilitating *At* in MWF [23], we were looking for compounds on which *Ct* could grow but that would be challenging for *At*. Our goal here is not to explain what we observed in MWF, but rather to design a comparable model chemical environment in which to study interactions in a more controlled way. LA was a good candidate, as it acted as a nutrient source for both species at low concentration, while at high concentration, it was toxic for *At* even though *Ct* could still grow well (Fig 1B). To generate quantitative predictions on the behavior of these 2 species in mono- and co-culture, we developed a mathematical model (ordinary differential equations (ODEs) describing dynamics of species and substrate abundances, see Methods) with parameters (e.g., growth and death rates) that best fit the mono-culture growth curves (Fig 1B). To capture the response of *At* to LA in the model, LA acted as a nutrient source that could also cause death at increasing concentrations (see Methods). We then used the model to predict how the 2 species are expected to interact if co-cultured at different LA concentrations (Fig 2A and 2B).

### Linoleic acid concentration determines interaction sign

The model predicted that increasing the concentration of LA in a co-culture of the 2 species could change the interaction sign from negative to positive. More specifically, at low concentration, both species compete for the sole nutrient source LA. Once the concentration becomes high enough to kill *At*, however, we expect to observe facilitation, as *Ct* consumes LA and reduces its concentration, making the environment less toxic for *At* (Fig 2A and 2B). We tested this prediction in the lab by growing *At* and *Ct* in mono- and co-culture in 0.1% and 0.75% LA as low and high LA concentrations, respectively. The results were in line with the predictions of the model: at low LA concentration (0.1%), *At* grew to significantly smaller population sizes in the co-culture compared to mono-culture (area under the growth curve (AUC) of *At* at 0.1% LA in mono-culture: $9\times10^8\pm4\times10^7$ versus co-culture: $7\times10^7\pm9\times10^6$, *t* test $P<0.001$, Fig 2C), showing that there was competition for LA that negatively affected the growth of *At*. At high LA concentration (0.75%), the presence of *Ct* in the co-culture rescued *At*, allowing it to grow to population sizes that were orders of magnitude greater than alone (AUC of *At* at 0.75% LA in mono-culture: $4\times10^7\pm2\times10^7$ versus co-culture: $2\times10^9\pm10^9$, *t* test $P = 0.03$, Fig 2D). The growth of *Ct* was not significantly affected by *At* in either condition (AUC of *Ct* in 0.1% LA in mono-culture: $9\times10^8\pm5\times10^7$ versus co-culture: $9\times10^8\pm10^8$, *t* test $P = 0.98$; AUC of *Ct* in 0.75% LA in mono-culture: $10^{10}\pm8\times10^9$ versus co-culture: $2\times10^{10}\pm2\times10^{10}$, *t* test $P = 0.42$, Fig 2E and 2F). Taken together, we classify the interactions between the 2 species as "ammensalism" at 0.1% LA and "commensalism" at 0.75% LA, with only 1 species being negatively or positively affected by the other, respectively [40].

Although these results matched the model predictions qualitatively, *At*'s growth in co-culture was greatly underestimated by the model (Fig 2B and 2D, green dashed lines). Furthermore, using the parameters estimated from all mono-cultures, the model does not correctly predict the hump-shaped growth of *At* at 0.75% LA (Fig 2D), even though it already assumes the accumulation of toxicity. This suggests that estimating model parameters where both growth and death are caused by a single compound is challenging. We next focused on

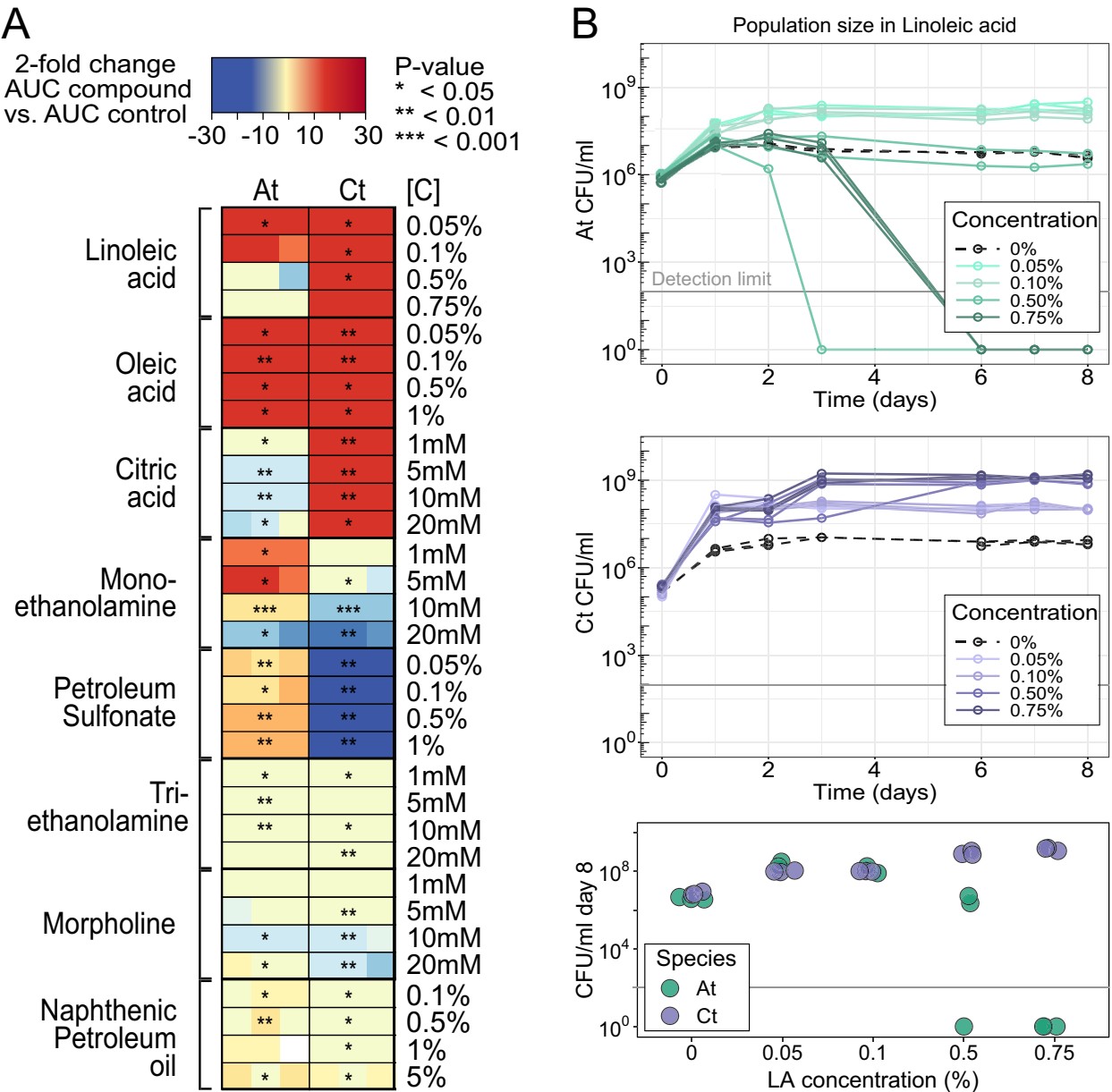

**Fig 1. Concentration-dependent effects of different compounds on the growth of *At* and *Ct*.** (A) Growth of *A. tumefaciens* (*At*) and *C. testosteroni* (*Ct*) in presence of different compounds at a range of concentrations in triplicates (each replicate is one colored square in the rectangle). The tested compound is the only carbon source added to the medium (see Methods). Heatmaps show the fold change between the AUC of each sample replicate and the AUC of the mean of the 3 control replicates where no compound was added. Blue shades represent negative fold change (bacteria grew to significantly smaller populations than the control) and orange shades represent positive fold change (bacteria grew to greater population sizes than the control). Statistical significance is determined by *t* tests comparing $N = 3$ AUCs at e.g., [C] = 0.05% to $N = 3$ AUCs at [C] = 0 (*: $P<0.05$,**: $P<0.01$,***: $P<0.001$). (B) Growth curves that generated the LA data in panel A. We observe some growth for both species in the absence of any carbon source (0%) but little competition when co-cultured (S1 Fig). While we were unable to find the source of this growth, similar observations have been previously reported [37–39]. All 3 technical replicates per condition are shown. The bottom panel shows population sizes at day 8 for a clearer comparison. The data underlying this figure can be found at https://zenodo.org/records/8033845. AUC, area under the growth curve; LA, linoleic acid.

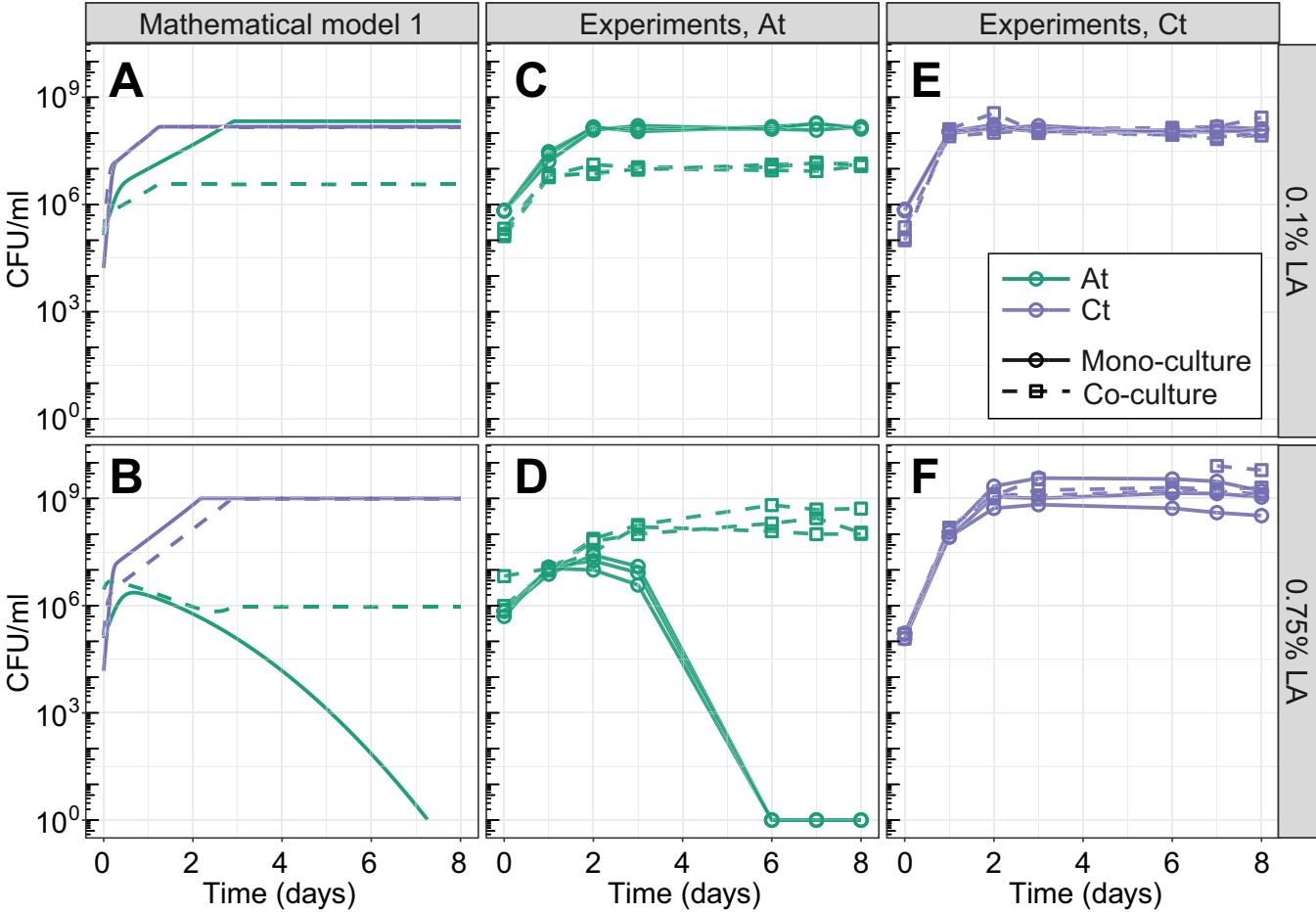

**Fig 2. Growth of the 2 species in mono- and co-cultures at the 2 LA concentrations, predicted by the model and in experiments.** (A, B) Predictions of model 1, where LA is a nutrient but becomes increasingly toxic over time. Model parameters were estimated by fitting to the mono-culture data (see Methods). (C–F) Experiments showing the growth of *At* (C, D) and *Ct* (E, F) at 0.1% LA (C, E) or 0.75% LA (D, F). Both species survive in mono-culture at 0.1% LA, whereas *At* grows and then dies in mono-culture at 0.75%. At 0.1% LA, *At* suffers from the presence of *Ct* in co-culture, while *Ct*'s growth is not significantly affected. At 0.75% LA, *At* is rescued by *Ct* in the co-culture, and *Ct*'s growth is not significantly affected. The model does a reasonable job at capturing the overall dynamics, but underestimates *At*'s growth in co-culture at 0.75% LA. For experimental data, all 3 technical replicates per condition are shown. See main text for statistics. The data underlying this figure can be found at https://zenodo.org/records/8033845. LA, linoleic acid.

exploring the mechanism behind this possible increase in toxicity, in particular, if we can decouple toxicity from the LA itself.

## ROS accumulates upon oxidation of linoleic acid and causes death of *At* unless *Ct* is present

Based on the literature, we hypothesized that spontaneous oxidation of LA might release ROS [41–44] even in the absence of bacteria. Accordingly, we used the thiobarbituric acid reactive substances (TBARSs) assay (see Methods) to test for the presence of ROS in our system [45]. Indeed, at both LA concentrations of the cell-free medium, ROS accumulated over time (Fig 3B, cell-free), supporting the idea of a chemical reaction leading to ROS production, such as spontaneous oxidation due to exposure of light and air [42,44]. ROS were significantly less abundant in *Ct* mono-culture and in the co-culture than in the mono-culture of *At* (*Ct* mono-culture versus *At* mono-culture, Tukey's test for multiple comparisons *P* = 0.0002; *Ct* co-culture versus *At* mono-culture, *P* = 0.0002, Fig 3B).

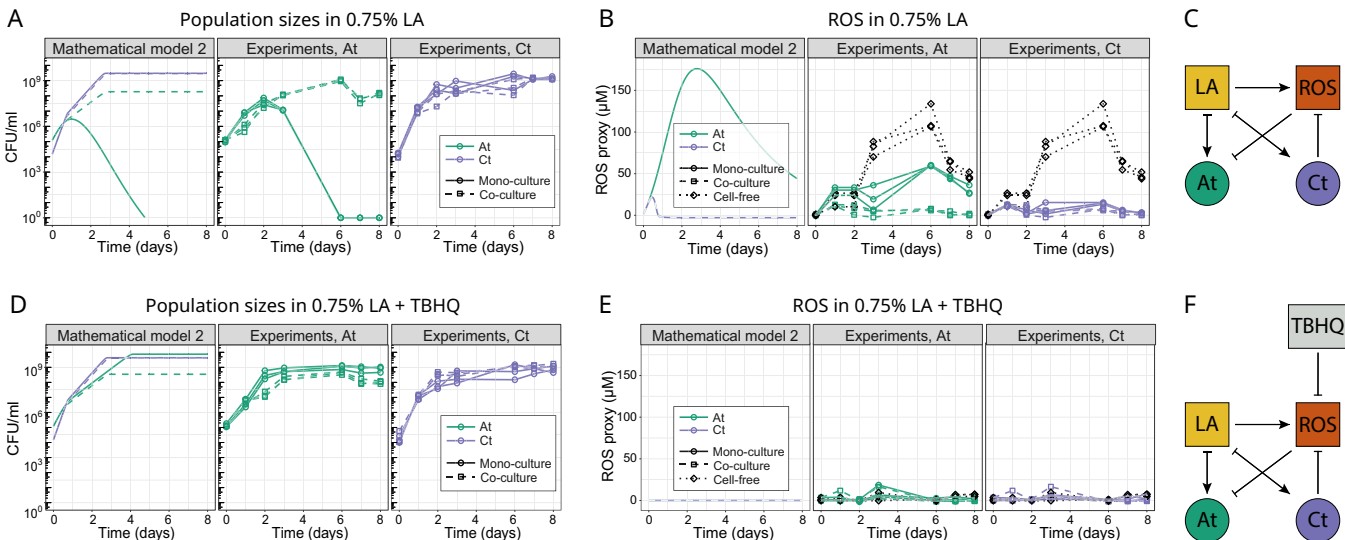

**Fig 3. Comparing population and ROS abundance over time.** (A, D) Population sizes of both species in mono- and co-culture at 0.75% LA (A) or 0.75% LA with the antioxidant TBHQ added daily (D) (see Methods). Panel (A) shows a biological repeat of the experiment shown in Fig 2, but where ROS was measured simultaneously (panel B). The mathematical model in the left panels of (A) and (D) is the second implementation of the model, where ROS is explicitly modeled as a separate chemical from LA. In model 2, LA is only a resource (see Methods). Model parameters were estimated by fitting to the mono-culture data (see Methods). (B, E) A proxy for ROS concentration (MDA-TBA2, see Methods for assay) over time in the different experimental treatments and in model 2 (left panel) in the 0.75% LA medium (B) or where TBHQ was added (E). The cell-free and co-culture data are identical in the *At* and *Ct* subpanels (in black), as they come from the same samples. For experimental data, all 3 technical replicates per condition are shown. In the model panels, the cell-free lines are hidden by the green ones for *At*, as they are identical. (C, F) A diagrammatic representation of the relationships between chemicals (in squares) and bacterial species (circles): (C) LA is consumed as a resource by both species and generates ROS. ROS inhibits *At* but can be reduced by *Ct*. (F) TBHQ inhibits ROS, making ROS removal by *Ct* superfluous. ROS removal by TBHQ or by *Ct* in co-cultures, rescues *At*. This translates into facilitation of *At* by *Ct* in the top row and competition in the bottom row, where TBHQ removes ROS. The data underlying this figure can be found at https://zenodo.org/records/8033845. LA, linoleic acid; ROS, reactive oxygen species; TBHQ, tert-butylhydroquinone.

The lack of ROS accumulation whenever *Ct* was present led us to hypothesize that *Ct* neutralizes ROS, reducing environmental toxicity and rescuing *At* in co-culture, allowing it to survive and grow (Fig 3A shows an experimental repeat of Fig 2D and 2F). These results are in line with the idea that toxicity was not caused by the increase of LA concentration itself, but rather by the accumulation of ROS. We adapted our model accordingly, leaving LA as a nutrient exclusively and adding ROS as an additional toxic compound that is generated through spontaneous LA oxidation. The co-culture growth prediction is qualitatively captured in this updated version of the model, in particular, the growth of *At* in high LA concentration is not underpredicted as it was in model 1 (Fig 3A compared to Fig 2B).

## Antioxidant rescues *At* and reverses the interaction from *Ct* to *At*

We used this second model to explore what would happen if we removed the toxicity by setting the production of ROS to 0 and leaving LA strictly as a nutrient. This led to 2 predictions: in absence of toxicity, (i) at 0.75% LA, *At* should survive even in mono-culture and reach a higher population size compared to 0.1% LA; and (ii) we should observe competition from *Ct* to *At* even at 0.75% LA (Fig 3D, left). To test these predictions, we first added an antioxidant molecule to the cell-free LA medium to verify whether its presence would decrease ROS concentration. We chose to use tert-butylhydroquinone (TBHQ) for its antioxidant properties [46] at a concentration of 1.5 μm that did not inhibit bacterial growth (S2 Fig). In addition to the cell-free medium, we also added TBHQ to *At* and *Ct* mono-cultures, and the co-culture at the beginning of the growth assay and every 24 h to have a regular input of fresh antioxidant,

mimicking continuous ROS neutralization by *Ct*. We found that TBHQ successfully decreased ROS concentration in all tested culture conditions compared to their value in standard 0.75% LA (Fig 3B and 3E). In support of prediction (i), adding TBHQ rescued *At* in mono-culture (Fig 3D, center), allowing it to reach a significantly higher population size in 0.75% LA + TBHQ compared to 0.1% (AUC of *At* mono-culture in 0.1% LA versus 0.75% LA + TBHQ, *t* test *P* = 0.013, compare solid lines in Fig 3D, center and Fig 2C). We suppose that in this ROS-free condition *At* could exploit the greater availability of LA as a nutrient. And as per prediction (ii), in 0.75% LA + TBHQ *At* grew to significantly lower population sizes in the presence of *Ct* than in mono-culture (Fig 3D, center, AUC of *At* mono-culture in 0.75% LA + TBHQ: $3.7 \times 10^9 \pm 1.3 \times 10^9$ versus co-culture: $7.9 \times 10^8 \pm 1.5 \times 10^8$, *df* = 5, *t* test *P* = 0.019), meaning that the interaction from *Ct* to *At* switched from facilitation to competition in the absence of toxicity. Overall, both model predictions were confirmed, demonstrating how we can shape the interaction from *Ct* to *At* by manipulating the level of toxicity in the environment.

## Separation of nutrient and toxic components in the model makes short-term coexistence between *At* and *Ct* more likely

In the first version of the model, toxicity depended on LA concentration, while in the second, it emerged on ROS accumulation, while LA acted exclusively as a nutrient. While this difference may seem like an implementation detail, we know from early theoretical work that 1 resource and 1 inhibitor are predicted to allow 2 species to coexist, while a single compound should only do so under very restrictive conditions [31,32]. We sought to verify this prediction and explore whether coexistence between *At* and *Ct* would be possible in either version of the model and in the experiments. We first extended the models to simulate a transfer experiment, where cultures were grown in batch for 72 h and then diluted 100-fold into fresh medium and regrown, and asked how long coexistence between the 2 species was possible. The second version of the model including ROS predicted a larger parameter range in which the 2 species could coexist in the short term (Figs 4A and S3). We tested this prediction experimentally by transferring 1% of the *At* and *Ct* mono-cultures and the co-culture in respective tubes every 72 h in both 0.1% and 0.75% LA. After 5 transfers, we found a significant variation for *Ct* co-culture in 0.1% LA compared to the beginning of the transfer experiment (first versus last transfer *Ct* co-culture 0.1% LA, *t* test, *P* = 0.0029), but there was no significant change for *At* in the same condition (first versus last transfer *At* co-culture 0.1% LA, *t* test, *P* = 0.30). Moreover, in 0.1% LA, competition was still evident from *Ct* to *At*, as *At* in mono-culture maintained a significantly higher population size than in the presence of *Ct* (mono-culture *At* 0.1% LA versus co-culture *At* 0.1% LA, *t* test *P*<0.0001, Fig 4B, left). This suggests that coexistence between *At* and *Ct* is possible in the short term despite the presence of negative interactions and provides further support for our updated model. In 0.75% LA, we observed the extinction of *At* mono-culture as we expected, but we found no significant variation neither for *At* nor for *Ct* in the co-culture (co-culture *At* 0.75%, *t* test *P* = 0.91, co-culture *Ct* 0.75%, *t* test *P* = 0.2, Fig 4B, right).

## The stress-gradient hypothesis holds in simulations predicting long-term dynamics

Although the time-scale of our experiments only allowed us to explore coexistence in the short term, it is still important to understand whether the 2 species would be able to coexist in the long term. We use our final model to explore this by simulating the outcome of mono- and co-cultures of *At* and *Ct* in a gradient of initial LA concentrations. We found no conditions where the long-term stable coexistence of the 2 species was possible, which is consistent with the idea

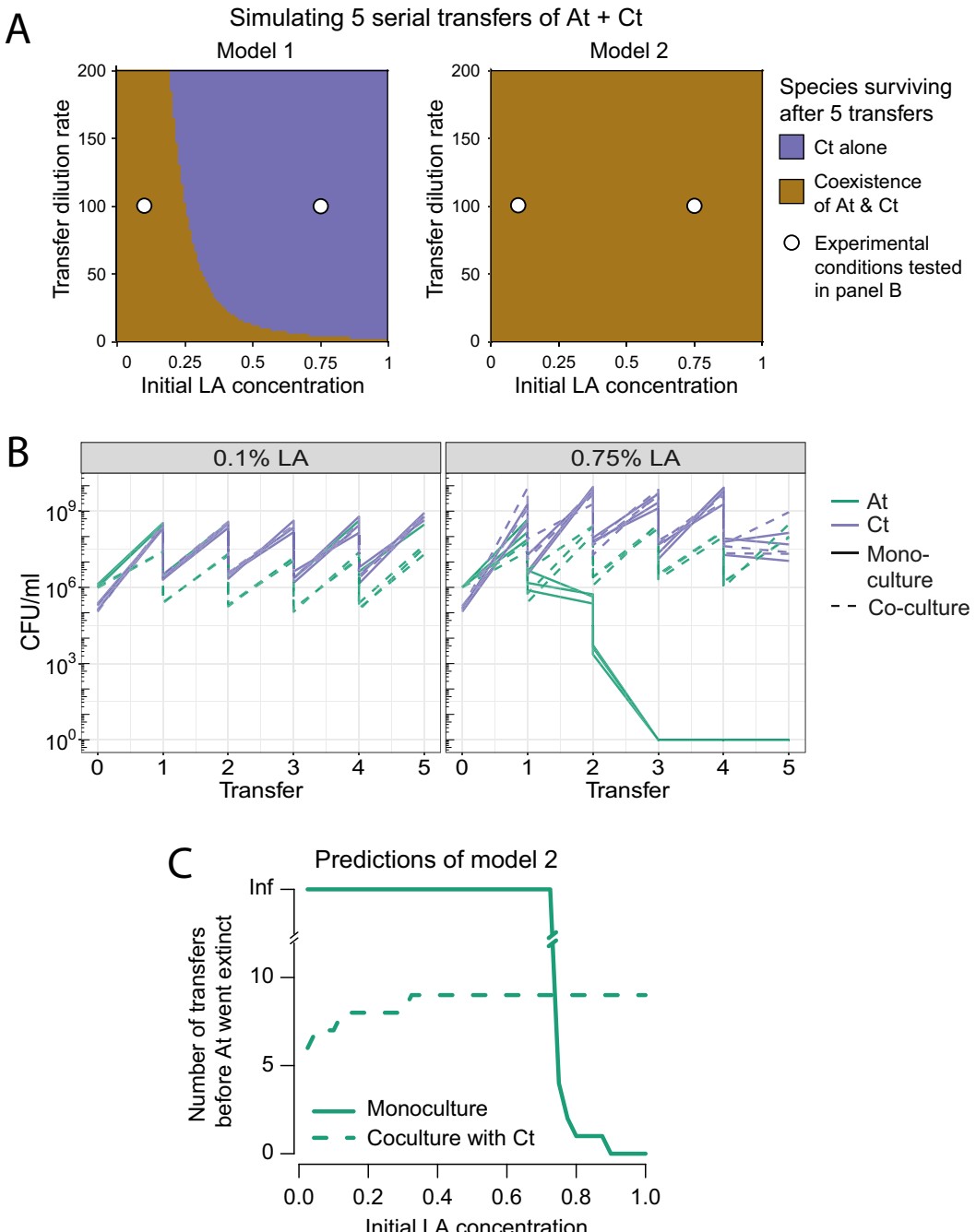

**Fig 4. Coexistence experiments and models.** (A) Prediction of short-term coexistence over 5 simulated serial transfers of the co-culture of *At* and *Ct* according to model 1 (left) and model 2 (right): Both models allow for short-term coexistence, but the parameter space in which this is possible is larger in model 2 (panel B, larger area representing coexistence compared to *Ct* surviving alone). White circles indicate the conditions in which experiments were run, as shown in panel B. (B) 5-Transfer experiment of *At* and *Ct* in mono- and co-culture at both 0.1% (left) and 0.75% LA (right). We show the population size in the initial culture ("transfer" 0), which is then quantified at each transfer (every 72 h). We illustrate the 100-fold dilution at each transfer, although this is not explicitly quantified. All 3 technical replicates per condition are shown. *At* mono-culture goes extinct as expected in 0.75% LA, but the 2 species coexist at both LA concentrations, as correctly predicted by model 2 in panel A. (C) Model 2 predicts that *At* should survive indefinitely in mono-culture at a 100-fold dilution rate up to 0.75% LA. In co-culture, *Ct* excludes it after a few transfers at low LA concentrations, but as the concentration increases, *At* can survive for longer in co- compared to mono-culture, meaning that *Ct* facilitates *At*'s survival by extending its duration (see also S3 Fig). The data underlying this figure can be found at https://zenodo.org/records/8033845. LA, linoleic acid.

that 2 species coexisting on a single resource is unlikely [47]. Nevertheless, to understand whether *Ct* had an effect on the extinction time of *At*, we measured the time (transfer) at which *At* went extinct when alone and compared it to its extinction time when in co-culture with *Ct*. At lower initial LA concentrations, *At* survives indefinitely in the long term in mono-culture (Fig 4C). As it eventually goes extinct in co-culture at all concentrations, the absence of coexistence at low concentration must be due to competitive exclusion by *Ct* (Fig 4C). In contrast, at higher initial LA concentrations, even though *At* quickly goes extinct in mono-culture, it survives for longer in the presence of *Ct*, meaning that *Ct* facilitates *At*'s survival. We then recapitulate the SGH in the long term: *Ct* competitively excludes *At* at low LA concentrations, but facilitates it by lengthening its survival at high LA concentrations.

## Discussion

Most current research on interactions between microbes has explored positive interactions through cross-feeding [5,10,14,48]. Our study instead focuses on positive interactions through detoxification [11,23] and shows how they can occur mechanistically. Facilitation through detoxification relates to the SGH. Here, with the help of our mathematical model, we show how the SGH can predict population dynamics: one species can provide a benefit to another by removing the stress that would otherwise drive the second species extinct, i.e., "niche facilitation" [24]. Experimentally removing the stress from the co-culture of the 2 species eliminates facilitation and restores competition (Fig 3D). This mechanistic understanding of the SGH increases its potential as a guiding principle for interspecies interactions [33,35].

How common is this phenomenon and is it specific to the compound we focused on, linoleic acid? Fig 1A shows that different compounds can be toxic in a species- and concentration-dependent way, and if species are also able to remove their toxic effect, they should behave similarly to the compound we chose to study. We also know that all aerobic microbes suffer from oxidative stress to some extent, as ROS is generated as a metabolic byproduct [49], and many frequently encounter ROS in their natural habitats [50]. In fact, ROS accumulation is a main plant stress response upon exposure to pathogens [51,52], and both organisms used in this study have been associated with plants [53,54]. One reason why the role of stress in mediating microbial interactions may not receive much attention, is that lab studies are typically designed to optimize bacterial growth, which a natural environment does not.

An important gap in our study is what type of ROS is accumulating, and how *Ct* is reducing the concentration of our ROS proxy: Is ROS being taken up and neutralized intracellularly or is *Ct* secreting extracellular enzymes that eliminate ROS? Answering this question is not as simple as analyzing *Ct*'s genome, since all aerobic bacteria are capable of dealing with oxidative stress and we find relevant enzymes in both *Ct* and *At* [55] (S1 Table). What we do know is that adding an antioxidant to the medium mimics the effects of *Ct* on *At*. Future experiments could explore the role of different types of ROS and different knockout mutants of *Ct* on the interaction between these 2 species. Spent-media experiments in a medium containing a simpler carbon source in addition to an ROS could also help to distinguish intra- and extracellular ROS elimination by *Ct*.

Our second main message is that even in very simple systems like the one we have studied here, several layers of interactions can play out simultaneously. Here, we find 2 layers: competition for the resource, linoleic acid, and facilitation through the removal of the toxin, ROS. Competition is present at all LA concentrations; if we remove environmental toxicity exogenously at high LA, the underlying resource competition is revealed. What we measure under given environmental conditions, then, is the net effect of species on each other, taking into account all the chemical substrates they may be competing for, feeding to one another or

removing to facilitate each other's growth. Because ROS generation is proportional to LA concentration, modifying the initial concentration of just that one compound alters the balance between resource abundance and toxicity, and thereby the dominance of competition or facilitation. This mechanism supports the intuition of the SGH (Fig 1 in [56]) that these interactions really are context dependent and do not simply emerge from confounding effects or methodological biases [57]. In this system, we have uncovered 2 mediators or environmental factors (LA and ROS); however, it is possible that there may be more that were not salient enough to be observed. More generally, microbes can construct a surprising number of new niches that could potentially mediate interactions with others [58,59].

Another important biological question is whether the underlying interactions can help to predict long-term coexistence. According to early theoretical work, coexistence on a single resource is only possible in small parameter regions in fluctuating environments, when metabolic trade-offs exist [47]. Intuitively, therefore, we did not expect coexistence in the competitive scenario at low LA concentration, but over the short time-scale of 5 transfers, our model predicts coexistence, which we recapitulate with the experiment.

It was less clear whether positively interacting species would coexist. Species that interact positively through cross-feeding are expected to coexist if the positive feedback is not too strong and leading to chaotic dynamics [24,30], but according to our model, 2 species that are competing for a single resource but facilitating each other through detoxification should not. Eventually, the stronger competitor should dominate. Nevertheless, we show that facilitation by detoxification can increase the duration of coexistence—again in line with the SGH—which may allow the weaker species to survive a limited period of harsh conditions [60,61]. It has been argued that the "expected time to extinction" is an appropriate measure of fitness [62]. Nevertheless, according to the model, in longer co-culture experiments, *At* should go extinct at all concentrations (Fig 4C), which would be an interesting hypothesis to test.

One indication that our model may be underestimating the potential for coexistence is that in Fig 4B, *At* does not appear to be close to extinction in either condition at transfer 5, while our model predicts that it should go extinct between transfers 6 and 10 (S3 Fig). This discrepancy leads us to suspect that despite our efforts to quantitatively predict outcomes of these co-cultures, our model is likely still missing further interaction layers between the 2 species like cross-feeding that were not observed in this study. Detecting cross-feeding in our system was challenging, as we found LA to not be easily amenable to standard chemical analysis methods. Understanding the coupling between detoxification and cross-feeding and their effect on long-term coexistence is left for future work.

This brings us to an important discussion point: How much detail is needed to construct a predictive model of a community like the one we have studied here? While we cannot give a definitive answer to this question, our study shows that details of the model matter. First, a consumer–resource (CR) model was needed to capture the context-dependency of interactions, compared to implicitly defining interactions as in a Lotka–Volterra framework [11,63,64]. Second, within the CR framework, whether we model a single compound whose effect changes or 2 compounds—one resource and one toxin—likely increases the parameter space under which the 2 species coexist (Fig 4A). This is in line with predictions that more environmental factors favor coexistence [28], and while we suspect that our model is still missing some of these factors to make it predictive in the long term, it nevertheless served to understand our system, how to control it, and to generate the testable hypothesis that toxicity may extend the duration of coexistence. On the one hand, this may be seen as bad news on fitting models to data: without a mechanistic understanding, it will be difficult to build a good predictive model. On the other hand, some underlying knowledge on the chemistry of the growth medium can go a long way. As long as parameterization is possible, the advantage of even a

partial mechanistic understanding is that it can generate more realistic hypotheses on the addition of further species or changes in resource concentrations, compared to an uninformed or abstract exploration of the parameter space of a model. Making sure enough interaction layers are included in the model to describe the observed experimental pattern requires a tight back and forth between models and data.

As we and others have shown, chemical compounds mediate positive or negative interactions between species, such that even in simple laboratory microcosms, several interaction layers may emerge whose balance and net effect will depend on initial conditions and how the chemical environment changes over time. Whether or not species will coexist in the long term will not only depend on the sign of these interactions but also the mechanism behind them: Are they removing stress for one another or feeding one another? Further verifying the model in this study and understanding how the layering of interactions plays out in the long term are next important steps.

## Methods

### Bacterial species and growth conditions

The species used in this study were *Agrobacterium tumefaciens* str. MWF001 (*At*) and *Comamonas testosteroni* str. MWF001 (*Ct*). More details on these strains can be found in [23] and their genome sequences on NCBI (Accession: PRJNA991498). We prepared separate overnight cultures in tryptic soy broth (TSB) starting from a single colony for each species. Cultures were incubated at 28˚C, shaken at 200 rpm. The day after, for each species, we measured the $OD_{600}$ (Ultrospec 10 cell density meter, Amersham Biosciences) and adjusted it to an $OD_{600}$ of 0.05 in 20 ml of fresh TSB, which we incubated for 3 h at the same conditions to reach exponential phase. We then measured the $OD_{600}$ to dilute cells to an $OD_{600}$ of 0.1 in 10 ml of minimal medium (MM, Table 1) in 15 ml tubes. We centrifuged these cell suspensions for 20 min at 4'000 rpm at 22˚C, discarded the supernatants and resuspended the pellets in 10 ml of PBS to remove any leftover TSB. We repeated this wash twice, resuspending the pellets in 10 ml of MM. Separately, we prepared 4 ml of the growth media in glass growth tubes, with 3 replicates per condition for every selected compound as indicated in Table 2. We aliquoted 40 μl of each bacterial culture (from the 10 ml suspended in MM) into the appropriate growth tube containing 4 ml of media to dilute bacteria at $10^5$–$10^6$ starting CFU/ml. Growth tubes were then incubated at 28˚C, 200 rpm for 8 days.

### Media preparation and compound storage

The compounds we used for media preparations were: linoleic acid, oleic acid, citric acid, monoethanolamine, petroleum sulfonate, triethanolamine, morpholine, and naphthenic petroleum oil. Previous studies show that bacteria are susceptible to benzotriazole and formaldehyde [65,66], so we chose concentrations above and below these thresholds. Petroleum sulfonate, naphthenic petroleum oil, linoleic acid, and oleic acid concentrations were chosen to resemble those commonly found in MWF [36]. Monoethanolamine, triethanolamine, citric acid, and morpholine concentrations were chosen to be similar to those already tested by the developers of the consortium [67]. Naphthenic petroleum oil and petroleum sulfonate were kindly given to us by Peter Kueenzi at Blaser Swisslube. All compounds were stored according to indications provided by the manufacturer: Linoleic and oleic acid were stored in individual 1 ml aliquots at −20˚C. Each aliquot was single-used to avoid multiple thaw–freeze processes, which could affect compound stability. We discarded and replaced aliquots that were more than 1 year old.

**Table 1. Preparation of MM.** The components listed in the table were mixed to prepare stock solutions. The stock solutions were combined according to the protocol to have the final MM at the top part of the table. $H_2O$up: "filled up to volume".

| Compound | Quantity |
|---|---|
| **Minimal medium (MM)** | |
| HMB 50X | 10 ml |
| M9 10X | 50 ml |
| ddH$_2$O | 440 ml |
| Final volume | 500 ml |
| **M9 10X** | |
| Na$_2$HPO$_4$ | 60 g |
| KH$_2$PO$_4$ | 30 g |
| NaCl | 5 g |
| NH$_4$Cl | 10 g |
| H$_2$Oup (final) | in 1 l |
| **HMB 50X** | |
| NTA (nitriolotriacetic acid) | 10 g |
| MgSO$_4$ * 7H$_2$O | 14.45 g |
| CaCl$_2$ * 2H$_2$O | 3.33 g |
| (NH$_4$)$_6$Mo$_7$O$_{24}$ * 4H$_2$O | 0.00974 g |
| FeSO$_4$ * 7H$_2$O | 0.099 g |
| Metals 44 | 50 ml |
| H$_2$Oup (final) | 1 l |
| **Metals 44** | |
| Na$_2$EDTA * 2H$_2$O | 0.387 g |
| ZnSO$_4$ * 7H$_2$O | 1.095 g |
| FeSO$_4$ * 7H$_2$O | 0.914 g |
| MnSO$_4$ * H$_2$O | 0.154 g |
| CuSO$_4$ * 5H$_2$O | 0.0392 g |
| Co(NO$_3$)$_2$ * 6H$_2$O | 0.0248 g |
| Na$_2$B$_4$O$_7$ * 10H$_2$O | 0.0177 g |
| H$_2$Oup | in 100 ml |

All growth media were prepared using a carbon-free minimal medium (Table 1), to which we added the appropriate concentrations of different compounds (or none) (Table 2). When possible, we prepared 50-fold concentrated stocks in water to standardize media preparation. Linoleic acid, oleic acid, petroleum sulfonate, and naphthenic petroleum oil are not fully miscible in water, and we used them as constantly shaken emulsions. This also prevented the preparation of stock solutions, and we therefore aliquoted the adequate amounts directly from their 99% pure stock (ddH$_2$O concentrations were adjusted accordingly as in Table 2). For linoleic acid and oleic acid, we thawed aliquots at room temperature before proceeding with media preparation. All assembled compound-supplemented media were incubated at 28˚C, 200 rpm for 3 h to allow complete mixing before distributing them into glass growth tubes and inoculating them with bacteria as described above. All media were prepared fresh on the morning of

**Table 2. Four different compound concentrations [C] plus a carbon-free medium were prepared by adding the compounds to MM (Table 1).** When possible, we prepared 50X concentrated stocks of the tested compounds to have a more standard procedure.

| | MM | MM + Compound [C] |
|---|---|---|
| **M9 10X** | 8 ml | 8 ml |
| **HMB 50X** | 1.6 ml | 1.6 ml |
| **Compound 50X [C]** | N/A | 1.6 ml |
| **ddH$_2$O** | 70.4 ml | 68.8 ml |
| **Final volume** | 80 ml | 80 ml |

each experiment, as we were aware that these compounds, particularly LA, could oxidize over time.

### Transfer experiments

We grew both *At* and *Ct* mono- and co-cultures for 72 h in both 0.1% LA and 0.75% LA. After 72 h, we transferred a 1% aliquot of each culture (40 µl) into 4 ml of fresh medium and grew bacteria for another cycle of 72 h. We performed 5 transfers, quantifying population sizes at each transfer as described next.

### Quantification of population size

To quantify bacterial population sizes over time, we took 20 µl aliquots from each growth tube, performed serial dilutions in 96-well plates filled with 180 µl of PBS and plated them on tryptic soy agar (TSA) plates or on lysogeny broth (LB) agar (depending on availability, no differences were observed). Plates were incubated at 28˚C. *Ct* and *At* formed countable colonies (CFUs) after 24 or 48 h of incubation, respectively. When growing in the co-culture, we could count *Ct* colonies because they appeared earlier (after 24 h). To count *At* colonies, we also plated the co-cultures onto LB agar supplemented with 14.25 µg/ml of sulfamethoxazole and 0.75 µg/ml of trimethoprim, on which only *At* could grow (native resistance). We also used a GFP marker integrated onto the chromosome of *At* to identify it when in doubt [23].

### Quantification of compound effect on bacteria

CFU counts were used to plot growth curves of CFU/ml over time and we calculated the AUC to have a better comparison between the different tested conditions and their MM control. We calculated the ratio between the AUC of each replicate per condition and the mean of the AUC of the 3 MM control replicates and used the $\log_2$-fold change of these data to build heat-maps showing the effect of each compound on each of the species (Fig 1). Raw data for all growth curves is shown in S4 Fig. *T* tests were performed to compare the tested conditions to the MM control.

### ROS detection assay

We used the TBARSs assay to indirectly assess the presence of ROS-induced oxidative stress as described in ref. [45]. Malondialdehyde (MDA) is the primary product of lipid peroxidation, the oxidative degradation induced by ROS. If there is ROS-induced degradation of LA in our media, this process would lead to MDA production. The TBARS assay measures the formation of the new adduct MDA-TBA2 upon reaction between the MDA in the medium and supplemented thiobarbituric acid (TBA). MDA-TBA2 presence is measured by its absorbance at 532 nm and the detected values are transformed in MDA-TBA2 concentrations through interpolation with a calibration curve built using 8 MDA-TBA2 standards at known concentrations (S5 Fig). All reacting solutions necessary for this assay were prepared following the detailed protocol in [45].

### Antioxidant assay

We used TBHQ as an antioxidant. After testing 4 different concentrations of TBHQ dissolved in dimethylsulfoxide (DMSO), we chose a concentration of 1.5 µm that had neither positive nor negative effects on the growth of the bacteria (S2 Fig) and rescued *At* in mono-culture at 0.75% (Fig 3D, center). TBHQ was prepared by dissolving it in 0.1% DMSO (below its toxic level [38]), as it was insoluble in water. We also verified that DMSO alone had no effect on

bacteria (S2 Fig). We prepared 800× concentrated stock of TBHQ+DMSO, such that only a small volume of 5 µl was added to our cultures every day. TBHQ+DMSO was prepared fresh every day to prevent degradation and loss of function of the antioxidant.

## Mathematical model

**Equations and fitting.**   We fitted mathematical models to the data from the experiments to determine whether species are expected to compete, facilitate each other, and whether they would coexist over long term serial transfers. Details are given in S1 Appendix, in particular in S2 Table we provide the description and units of the different parameters and state variables used in the models. All code is available at https://zenodo.org/records/10396269.

In the first model, bacterial species abundance $B$ over time depends on the concentration of LA $C$ according to its consumption through a Monod uptake function, with maximum growth rate $r_C$, half-saturation constant $K_C$, and yield $Y_C$. LA also induces mortality depending on its concentration. We assumed that its toxicity increases linearly over time and is proportional to the LA concentration, leading to the linear expression $(\beta + \gamma t)C$. This linear increase may appear arbitrary, but it was needed to get a reasonable fit to the mono-culture of At at 0.75% LA, where the population grows initially and then drops abruptly (see S6 Fig and S3 Table). LA concentration $C$ varies only due to consumption by bacteria. This formulation is similar to the classic growth-inhibiting substrate approach as described in example 1 from [24] with a hump-shaped functional response (Haldane or Type IV), except that it allows for a negative growth rate (death) while the Haldane form tends to 0. The equations for the variation of bacterial abundance and LA concentration in a mono-culture are as follows:

$$\frac{\mathrm{d}B}{\mathrm{d}t} = \frac{rCB}{C + K} - (\beta + \gamma t)CB \tag{1}$$

$$\frac{\mathrm{d}C}{\mathrm{d}t} = -\frac{1}{Y}\frac{rCB}{C + K} \tag{2}$$

The equations for 2 species $B_1$ and $B_2$ in co-culture in LA are given in S1 Appendix. We used this model to fit the growth of At and Ct in mono-culture in a range of concentrations of LA (0.05%, 0.1%, 0.5%, and 0.75%) and compared the predictions for the abundances of the 2 species to the experimental data in a short-term co-culture and their long-term coexistence over transfers. In this first model, the estimated parameters were $r_{C1}$, $r_{C2}$, $Y_{C1}$, $Y_{C2}$, $K_{C1}$, $K_{C1}$ and the toxicity parameters $\beta_1$, $\beta_2$, $\gamma_1$, $\gamma_2$. The best-fit estimates are listed in S2 Table.

We then used a second model that accounts for the production of ROS by LA oxidation. The equations for the mono-culture growth are now:

$$\frac{\mathrm{d}B}{\mathrm{d}t} = \frac{rCB}{C + K} - \beta RC \tag{3}$$

$$\frac{\mathrm{d}C}{\mathrm{d}t} = -\frac{1}{Y}\frac{rCB}{C + K} - \frac{1}{m}(d + eR)C \tag{4}$$

$$\frac{\mathrm{d}R}{\mathrm{d}t} = (d + eR)C - lR - \alpha BR \tag{5}$$

Because we had acquired data on the spontaneous oxidation of LA in cell-free media, we could first estimate the parameters $d$, $e$, $m$, and $l$ using an ROS proxy measured at different LA concentrations (S7 Fig). We then fixed these parameters as estimated from the mono-culture

and further estimated the parameters of growth, toxicity, and detoxification for each species. All parameter estimations were obtained using the modFit function from FME package (version 1.3.6.1) in R version 4.1.0, which uses the Levenberg–Marquardt algorithm from the nls. lm function (min.pack package). The objective function was defined by the $\log_{10}$ of the error between data points and the fit (both for fitting CFU data or ROS data). Initial parameter sets were obtained by fitting the data by trial and error. Parameter values were explored linearly or on a log scale, which led to similar estimates. In the linear exploration, an upper bound was set to $10^{10}$ and a lower bound to 0. In the main text, we chose one arbitrary parameter set for each model, which are not intended as being the absolute best-fit, but more as a representative of the output from the fitting routine.

## Comparing model predictions to co-cultures and transfers

For both models, we used the parameters obtained from the estimation of the mono-culture data to predict co-culture dynamics and compared them to the actual co-culture data (see equations 11–14 for co-culture dynamics in S1 Appendix). In S3 Table, we present some quantitative measures for the performance of the models. In particular, the goodness-of-prediction measure allows us to compare the prediction ability of the model by computing the error between the co-culture growth predicted by fitting the mono-culture data and the actual co-culture data. The goodness-of-fit measure allows us to compare different versions of model 1 or of model 2 to assess whether to add complexity to the model. The goodness-of-fit measures of model 1 and model 2 are not directly comparable, firstly because the mono-culture fitting routine in model 1 uses 4 concentrations of LA, while the fitting of model 2 uses only 2; secondly, in model 2, the ROS dynamics are included in the fitting routine, which may explain why the overall goodness-of-fit (based on the CFU data) is better in model 1 than in model 2.

We also simulated the serial transfers with varying dilution rates and initial LA concentrations to predict the likelihood of coexistence of the 2 species over time. The transfer parameter sweeps were coded in C++. In the second model, we mimicked the addition of an ROS quencher to the media by setting initial ROS concentration to zero, as well as parameters *d*, *e*, and *l* and compared the predicted dynamics to the actual data using TBHQ (figure not shown).

## Supporting information

**S1 Appendix. Details on the modeling and fitting analysis.** Descriptions of models 1 with implicit toxicity and model 2 with explicit toxicity due to ROS. All code is available at https://zenodo.org/records/10396269.
(PDF)

**S1 Fig. Mono- and co-cultures in 0% LA.** Growth in MM of *At* (top) and *Ct* (bottom) in the presence (dashed lines) and absence (solid lines) of the other species. We find that in all cases, bacteria grow in MM where no external carbon source has been added. *Ct*'s population size is only slightly inhibited by the presence of *At* in MM, while *At* grows similarly whether *Ct* is present or not. These data show that the competitive effect of *Ct* on *At* arises when 0.1% LA is added to MM, which is also shown on the plots. We argue then, that despite significant growth in MM, it is legitimate to focus on LA being the carbon source that is mediating the competitive effect. The data underlying this figure can be found at https://zenodo.org/records/8033845.
(PDF)

**S2 Fig. Effect of TBHQ at different concentrations on bacteria.** We tested the effect of 4 different concentrations of TBHQ (150 μm, 15 μm, 1.5 μm, and 0.375 μm) on each of the species in mono- and co-culture (A–D) and compared it to the MM alone (no TBHQ). We also tested the effect of DMSO, the solvent we used to prepare TBHQ, on growth (F). Based on this, we used MM + 1.5 μm as our antioxidant. The data underlying this figure can be found at https://zenodo.org/records/8033845.
(PDF)

**S3 Fig. Simulation of transfer trajectories.** Transfer trajectories are simulated to predict the long-term coexistence of *At* and *Ct* according to the model. Here, we show for model 2, transfer of mono-cultures or co-cultures in the 2 LA conditions, with a dilution rate of 100. Even though after 5 transfers the 2 species are coexisting (above an extinction threshold), it is clear that this coexistence is not stable as *At* density is declining. Code that generated this figure is available at https://zenodo.org/records/10396269.
(PDF)

**S4 Fig. Growth curves of *At* and *Ct* in MWF compounds.** We chose compounds representative of MWF composition and grew *At* and *Ct* in increasing concentrations of the following compounds: petroleum sulfonate (A), naphthenic petroleum oil (B), triethanolamine (C), monoethanolamine (D), citric acid (E), morpholine (F), and oleic acid (G). Darker gradient of blue curves represents increasing compound concentration.
(PDF)

**S5 Fig. Calibration curve ROS quantification.** Malondialdehyde bis(dimethyl acetal) (MDA) concentration is used as a proxy for ROS accumulation: the higher the MDA concentration, the higher is the ROS abundance in the sample [45]. We chose 6 increasing concentrations of MDA (0, 2.5, 5, 10, 20, 40, 80, 160 μm) and we used 3 replicates per concentration to build a calibration curve. We performed the TBARS assay on each calibration sample as described in [45] and measured the 532 nm absorbance. We subtracted the absorbance of the blank (0 μm MDA) from each calibration sample and plotted MDA concentration versus blank-subtracted absorbance. We obtained a calibration curve and used its parameters to calculate the MDA concentration of the experimental samples shown in Fig 3. The data underlying this figure can be found at https://zenodo.org/records/8033845.
(PDF)

**S6 Fig. Model 1 fitting without toxicity accumulation.** We consider a simpler model in which there is no toxicity accumulation ($\gamma = 0$ in Eq 1). LA is then directly toxic for *At* and the initial growth results from the consumption of the nutrients in the minimal medium. This model also captures the dynamics in mono-culture (panels A and B) and gives a qualitatively similar prediction for the co-cultures (panels C and D) as in the model with toxicity accumulation, but the error is larger because the growth of *At* is even more underestimated compared to the data (see S3 Table). The data underlying this figure can be found at https://zenodo.org/records/8033845 and code used for fitting is available at https://zenodo.org/records/10396269.
(PDF)

**S7 Fig. ROS cell-free parameter fit.** We used the MDA concentration in the cell-free media at both LA concentrations to estimate the intrinsic ROS parameters in the absence of bacteria. ROS concentration (in arbitrary units) first increases then stabilizes. The data underlying this figure can be found at https://zenodo.org/records/8033845 and code used for fitting is available at https://zenodo.org/records/10396269.
(PDF)

**S8 Fig. Fitting the growth of *At* and *Ct* in 0% LA.** Since the nutrient that allows *At* and *Ct* to grow in the MM is unknown, we set it to an arbitrary concentration of 0.01 to fit the parameters of a consumer–resource model using both mono- and co-cultures of *At* and *Ct*. Here, the data used to fit the model are from the second experiment with ROS measurements. The model prediction is shown in the thick transparent lines, with solid lines for mono-cultures and dashed lines for co-cultures. The data underlying this figure can be found at https://zenodo.org/records/8033845 and code used for fitting is available at https://zenodo.org/records/10396269.
(PDF)

**S9 Fig. Model 2 fitting with linear ROS dynamics.** We consider a simpler model in which there is no ROS-induced oxidation of LA which removes the nontrivial feedback loop ($e = 0$ in Eqs 4 and 5). This model is quite similar to model 2 with the ROS-induced oxidation term but has a higher error both in the mono-culture fit and in terms of prediction of the co-culture growth (see S3 Table). Raw data are available at https://zenodo.org/records/8033845 and code used for fitting is available at https://zenodo.org/records/10396269.
(PDF)

**S1 Table. ROS resistance genes in *At* and *Ct*.** We searched for a list of putative ROS-degrading enzymes from a recent review paper [54] by searching the annotated genes of our 2 strains, based on sequences available on NCBI (Accession: PRJNA991498). We show the gene families listed in [54] and whether we found genes of the same family in our 2 genomes with gene number shown in brackets. This analysis shows that gene presence/absence tells us little about which of the 2 strains is more resistant to ROS.
(PDF)

**S2 Table. Description of state variables and parameters used in the models, with the estimates from fitting on mono-cultures.** The compounds (LA, ROS, minimal medium nutrient) concentration unit is arbitrary (au). Code used for fitting is available at https://zenodo.org/records/10396269.
(PDF)

**S3 Table. Sum of squared-errors between the model and the data for the different models.** We compute the squared errors (in $\log_{10}$) between the model ODE simulations and the data points to get a proxy for the goodness of the fit. We sum the errors for *At* and *Ct* growth at 0.1% and 0.75% LA to obtain a global goodness-of-fit proxy: for example, the goodness-of-fit of model 1 is 189.7. Two goodness-of-fit measures are considered: goodness-of-fit which is calculated using the mono-culture data and goodness-of-prediction which is calculated using the co-culture data. Model 1 and model 2 are the models presented in the main text (implicit toxicity and ROS-driven toxicity, respectively). Model 1 with no toxicity accumulation and model 2 with linear ROS dynamics are simpler versions of these models in which we ignore nontrivial dynamics such as the positive feedback loop in ROS generation ($e = 0$) and the accumulation of toxicity over time in model 1 ($\gamma = 0$). Model 1 has better goodness-of-fit measures for the mono-culture growth compared to model 2—this comparison must be taken with caution since model 2 also fits the ROS dynamics at the same time, and only uses 2 LA conditions where in model 1; 4 concentrations of LA were used to fit the mono-cultures. In terms of predicting the co-culture dynamics, model 2 with nonlinear ROS dynamics provides the most accurate prediction for the co-culture data. All code is available at https://zenodo.org/records/10396269.
(PDF)

## Acknowledgments

We thank Andrew Quinn and Philipp Engel for their help to obtain metabolomics data that are not included in the manuscript, and Katia Annen for help with the experiments. We thank Peter Kueenzi at Blaser Swisslube for the chemical compounds used to generate the data in Fig 1. We thank Afra Salazar de Dios, Julien Luneau, Margaret Vogel, Oliver Meacock, and Massimo Amicone for insightful comments on the manuscript.

## Author Contributions

**Conceptualization:** Rita Di Martino, Aurore Picot, Sara Mitri.

**Data curation:** Rita Di Martino.

**Formal analysis:** Rita Di Martino, Aurore Picot, Sara Mitri.

**Funding acquisition:** Sara Mitri.

**Investigation:** Rita Di Martino, Aurore Picot, Sara Mitri.

**Methodology:** Rita Di Martino, Aurore Picot, Sara Mitri.

**Project administration:** Sara Mitri.

**Resources:** Rita Di Martino, Aurore Picot, Sara Mitri.

**Software:** Rita Di Martino, Aurore Picot, Sara Mitri.

**Supervision:** Sara Mitri.

**Validation:** Rita Di Martino, Aurore Picot, Sara Mitri.

**Visualization:** Rita Di Martino, Aurore Picot, Sara Mitri.

**Writing – original draft:** Rita Di Martino, Aurore Picot, Sara Mitri.

**Writing – review & editing:** Rita Di Martino, Aurore Picot, Sara Mitri.

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
