## [Editor Report · Decision Letter 0]

6 Jun 2023

Dear Dr Mitri, 

Thank you for submitting your manuscript entitled "Oxidative stress changes interactions between two bacterial species" for consideration as a Research Article by PLOS Biology.

Your manuscript has now been evaluated by the PLOS Biology editorial staff as well as by an academic editor with relevant expertise and I am writing to let you know that we would like to send your submission out for external peer review.

Once your full submission is complete, your paper will undergo a series of checks in preparation for peer review. After your manuscript has passed the checks it will be sent out for review. To provide the metadata for your submission, please Login to Editorial Manager (https://www.editorialmanager.com/pbiology) within two working days, i.e. by Jun 08 2023 11:59PM.

Kind regards,

Luke 

Lucas Smith, PhD

Senior Editor

PLOS Biology 

lsmith@plos.org

on behalf of my colleague Paula (who is the handling editor for your study, but who is out of the office this week)

Senior Editor

PLOS Biology

---

## [Decision Letter · Decision Letter 1]

24 Jul 2023

Dear Dr. Mitri,

Thank you for your patience while your manuscript "Oxidative stress changes interactions between two bacterial species" was peer-reviewed at PLOS Biology. It has now been evaluated by the PLOS Biology editors, an Academic Editor with relevant expertise, and by several independent reviewers. 

In light of the reviews, which you will find at the end of this email, we would like to invite you to revise the work to thoroughly address the reviewers' reports.

As you will see below, the reviewers think that the work is interesting, but they all have some issues that would need to be solved before further consideration. In particular, reviewers #1 and #2 ask for more clarification regarding the results on long-term dynamics and coexistence. Please address all the reviewers' concerns. 

Given the extent of revision needed, we cannot make a decision about publication until we have seen the revised manuscript and your response to the reviewers' comments. Your revised manuscript is likely to be sent for further evaluation by all or a subset of the reviewers.

**IMPORTANT - SUBMITTING YOUR REVISION**

*Re-submission Checklist*

*Published Peer Review*

*PLOS Data Policy*

*Blot and Gel Data Policy*

Sincerely,

Paula

---

Senior Editor

PLOS Biology

REVIEWS:

Reviewer #1: Jeff Gore. Interactions between bacteria in microbial communities.

Reviewer #2: Evolution of microbial communities.

Reviewer #3: Djordje Bajic. Evolution of microbial communities.

Reviewer #1: Understanding how species interactions depend upon the environment is a central question in ecology. One pattern that has been observed is that species interactions often become more positive when the environment is more challenging for the species (the so-called stress gradient hypothesis (SGH)). Here the authors study a co-culture of Agrobacterium tumefaciens (henceforth At) and Comamonas testosteroni (Ct) growing in a defined medium containing linoleic acid (LA) as the sole carbon source. The authors find that this carbon source becomes toxic for At at high concentration due to the accumulation of Reactive Oxygen Species (ROS). Using a combination of experiment and modeling, the authors explain how the interactions in the two-species co-culture move towards more competition (or facilitation) by reducing (or increasing) the initial LA concentration, consistent with expectations from the SGH. In addition, they find that adding an antioxidant reduces environmental toxicity, thus moving the interaction back to competition. I very much enjoyed the clear writing and the elucidation of a mechanistic origin behind how the environment can determine the sign of interactions between species, and I have only one question to be addressed (I will also note that Sara Mitri's papers are unusually carefully edited, as a typical submitted paper contains a dozen typos, whereas I didn't notice any in this paper nor in some other papers of hers that I have reviewed…. Thanks to the authors for putting in this extra effort).

Long-term dynamics (line 194): "We found no conditions where the long-term stable coexistence of the two species was possible." It seems to me that At in the co-culture experiments is surviving for a pretty long time, and there is no evident decrease in the cell density over five cycles. I would have interpreted this as reasonable evidence for "long-term" survival, so from that standpoint I would say that Model 2 is still not consistent with the experimental data. (Of course, the Ct population density dips on the final cycle, and this could have unpredictable effects for the co-culture…. However, this is probably just due to experimental variation on that cycle?). Do the authors really think that there co-culture would not survive indefinitely? I suspect that there are some experimental conditions where they would… In particular, cross-feeding is such a ubiquitous phenomenon that most single carbon sources can support dozens of species / strains. Obviously, if you write down a resource explicit model that doesn't include cross-feeding then you will predict that only a single species will survive, but that is because the model has assumed that there is no cross-feeding. The model that the authors develop is clearly capturing some essential dynamics that are present within the community, but it is still making a strong assumption about the lack of cross-feeding, and without clear evidence of competitive exclusion at high resource concentration it seems odd to end the paper by making a strong modeling prediction that may or may not be justified.

Reviewer #3: This study builds up on previous work by Dr. Mitri's group, studying how the ecology of microbial communities changes in response to harsher versus more benign environments. Here, they provide a detailed mechanistic characterization of the SGH hypothesis in their system, quantitatively establishing how the concentration of a single compound can modulate the sign of the ecological interaction through its associated toxicity. I find it quite brave to work with such a difficult system, and being able to disentangle it mechanistically has a lot of merit. It is also a beautiful example of how the back and forth between models and experiments can help mechanistically characterize the observed interactions and population dynamics. I also greatly appreciated the depth and breadth of the discussion, including the generalizability of the phenomenon, the multi-layered nature of even apparently simple microbial interactions, and the level of mechanistic detail that we need to include in our models. The findings are novel, the paper is very well written, the questions are answered rigorously and the limitations are discussed transparently. Thus, I am overall very enthusiastic with this contribution whose importance spans from community ecology in general to microbial ecology and microbiology. 

Minor points 

I can't help commenting on the growth with no added carbon source in Fig. 1. I acknowledge the authors transparency and care in documenting it with references. We have also noticed a similar phenomenon in our experiments, but it remains striking to me. Do authors know if perhaps these species are forming any storage molecule during "feast" times, e.g. during the precultures in TSB, that they are then using during the "famine" period?

Statistical analysis is adequate throughout the paper. However, please clarify the statistical analysis reported in caption of Fig 1. If I understand correctly, in each case the t test was performed between a condition (compound X added at concentration Y, three replicates) and all the replicates of the condition with no added compound (how many?). Please report the N as well. 

L355 - Could you please clarify how these compounds added to LB help distinguish the two species, or provide reference.

---

## [Decision Letter · Decision Letter 2]

7 Dec 2023

Dear Dr Mitri,

Thank you for your patience while we considered your revised manuscript "Oxidative stress changes interactions between two bacterial species" for publication as a Research Article at PLOS Biology. Please note that I have taken over as the handling editor of your manuscript since Paula Jauregui has now recently moved on from PLOS Biology. This revised version of your manuscript has been evaluated by the PLOS Biology editors, the Academic Editor and the original reviewers.

Based on the reviews, I am pleased to say that we are likely to accept this manuscript for publication, provided you satisfactorily address the following data and other policy-related requests that I have provided below (A-E):

(A) We would like to suggest the following modification to the title: 

“Oxidative stress changes interactions between two bacterial species from competitive to facilitative”

(B) You may be aware of the PLOS Data Policy, which requires that all data be made available without restriction: http://journals.plos.org/plosbiology/s/data-availability. For more information, please also see this editorial: http://dx.doi.org/10.1371/journal.pbio.1001797

Thank you for already providing the individual numerical values that underlie the summary data displayed in the figure panels in the Raw Data. In the Data Availability Statement, I note that it is your intention to deposit this data in the Zenodo repository but preferred to wait until you had the final data. I would be grateful if you could now deposit the data for the following figures during this round of revision:

Figure 1A-B, 2C-F, 3A-B, 3D-E, 4A-C, S1, S2, S3, S4A-G, S5, S6A-D, S7, S8, S9

(C) Please also ensure that each of the relevant figure legends in your manuscript include information on *WHERE THE UNDERLYING DATA CAN BE FOUND*, and ensure your supplemental data file/s has a legend.

(D) Please ensure that your Data Statement in the submission system accurately describes where your data can be found and is in final format, as it will be published as written there. 

(E) Please note that per journal policy, the specific names of the two bacterial species studied should be clearly stated in the abstract of your manuscript. 

We expect to receive your revised manuscript within two weeks. 

*Published Peer Review History*

*Press*

Kind regards,

Richard

Richard Hodge, PhD

rhodge@plos.org

Reviewer remarks:

Reviewer #1 (Jeff Gore, signs review): In my opinion, the authors have addressed the questions and concerns expressed by me and the other referees, and the manuscript should be published as it is. As discussed by the other referees (eg "strategic" vs "predictive" models), the relationship between experiments and models is subtle and there are a variety of different approaches within our community. I might have phrased things differently, but I do not believe that we should be too prescriptive as referees. On the topic of growth in the absence of added carbon, we too have observed this and have (perhaps not surprisingly...) found it difficult to characterize. 

Reviewer #2: I've reviewed the revised manuscript and the authors' response. 

The authors have nicely addressed all of my (and other reviewers') comments and I feel that the paper is now acceptable for publication.

Reviewer #3 (Djordje Bajic, signs review): The authors addressed my comments more than sufficiently (even if they were quite minor). I think the paper is now acceptable for publication and represents an excellent contribution.

---

## [Editor Report · Decision Letter 3]

22 Dec 2023

Dear Dr Mitri,

On behalf of my colleagues and the Academic Editor, Tobias Bollenbach, I am pleased to say that we can in principle accept your manuscript for publication, provided you address any remaining formatting and reporting issues. These will be detailed in an email you should receive within 2-3 business days from our colleagues in the journal operations team; no action is required from you until then. Please note that we will not be able to formally accept your manuscript and schedule it for publication until you have completed any requested changes.

*IMPORTANT* 

During the production process and proofs, I would be grateful if you could please state in the figure legends (both main and supplementary) where the underlying data can be found for the figure panels presented (i.e. the Zenodo repository, providing the relevant DOI). In addition, I noticed that the figure legends currently contain a question mark for the individual figure numbers, so I would be grateful if this error could be fixed at this stage.

PRESS

Best wishes, 

Richard

Richard Hodge, PhD

rhodge@plos.org

PLOS
